# Genetic Diversity in the Diminazene Resistance-Associated P2 Adenosine Transporter-1 (*AT-1*) Gene of *Trypanosoma evansi*

**DOI:** 10.3390/ani15050756

**Published:** 2025-03-06

**Authors:** Shoaib Ashraf, Ghulam Yasein, Qasim Ali, Kiran Afshan, Martha Betson, Neil Sargison, Umer Chaudhry

**Affiliations:** 1College of Veterinary Sciences, Riphah International University, Lahore 54660, Punjab, Pakistan; shoaib.ashraf@ucalgary.ca; 2Cumming School of Medicine, University of Calgary, Calgary, AB T2N 1N4, Canada; 3Department of Parasitology, University of Veterinary and Animal Sciences Lahore, Lahore 54000, Punjab, Pakistan; yasein2005@gmail.com; 4Department of Parasitology, University of Agriculture D. I. Khan, Dera Ismail Khan Khyber 29050, Pakhtunkhwa, Pakistan; qasim8485@gmail.com; 5Department of Zoology, Faculty of Biological Sciences, Quaid-i-Azam University, Islamabad 45320, Punjab, Pakistan; kafshan@qau.edu.pk; 6School of Veterinary Medicine, University of Surrey, Guildford GU2 7XH, UK; m.betson@surrey.ac.uk; 7Royal (Dick) School of Veterinary Studies, University of Edinburgh, Edinburgh EH8 9JU, UK; neil.sargison@ed.ac.uk; 8Lewyt College of Veterinary Medicine, Long Island University, New York, NY 11548, USA

**Keywords:** diminazene, *Trypanosoma evansi*, P2 adenosine transporter-1

## Abstract

*Trypanosoma evansi* is a highly pathogenic bloodborne protozoan parasite of many animals. It is widespread in South Asia, North Africa, and the Middle East. It has also been exported to Australia, Latin America, and Europe through animal movement. The underlying disease causes high production losses and mortality in affected livestock species. Diminazene is consistently used to treat *T. evansi*, and resistance to this drug is an emerging threat in ruminant livestock (such as sheep, goats, and cows) and camels, resulting in huge economic losses. The present study aimed to develop a high-throughput strategy based on molecular approaches to improve the understanding of diminazene resistance in *T. evansi*.

## 1. Introduction

*Trypanosoma evansi* is a highly pathogenic bloodborne protozoan parasite of a wide range of animals that causes the disease known as Surra [1]. *T. evansi* is widespread in South Asia, North Africa, and the Middle East. *Trypanosoma brucei* predominates outside the African tsetse fly belt and causes African sleeping sickness. Both species have been exported through animal movement to Australia, Latin America, and Europe [2]. *Tabanus* and *Stomoxys* flies transmit the disease and act as a reservoir between different animal hosts [3]. Trypanosomiasis causes huge production losses, high morbidity, and mortality in the affected livestock species.

Surra is usually treated with antiprotozoal drugs such as diminazene, which was first introduced in 1955 [4]. The P2 adenosine transporter-1 (*AT-1*) protein is required for the uptake of diminazene in *Trypanosoma* spp. [5]. Once diminazene is taken up, it enters the mitochondria, where it binds to a minor groove of kDNA. By doing so, the drug induces changes in DNA topology and inhibits the topoisomerase enzyme. Consequently, the functioning of the mitochondrion is impaired as DNA is cleaved [6].

Diminazene resistance in trypanosomes is an emerging threat in ruminant livestock species and camels causing staggering annual losses of over PKR one billion due to trypanosomiasis. Even though resistance to diminazene has emerged, it is unlikely that new alternative drugs will be developed in the near future. In *Trypanosoma brucei gambiense*, several studies have demonstrated a functional relationship between diminazene resistance and the inhibition of the P2 adenosine transporter-1 (*AT-1*) locus [7]. Diminazene resistance-associated P2 adenosine transporter-1 (*AT-1*) mutations have been reported in at least seven codons (71, 178, 181, 239, 286, and 380) of *Trypanosoma brucei gambiense* [8,9,10,11]. The current understanding of *T. evansi* genetic diversity at the *AT-1* locus is limited, and potential links between mutations in this transporter and diminazene resistance remain elusive. As *Trypanosoma brucei gambiense* and *Trypanosoma evansi* are closely related species, it is plausible that mutations at this locus may also contribute to diminazene resistance in *T. evansi.* In this context, the role of this locus in the resistance to diminazene in *T. evansi* is not fully understood [11,12,13]. Genetic diversity has been described in protozoan parasites. Studies have reaffirmed the contribution of high parasitic population size to genetic diversity strongly linked with mutations in cytochrome b and buparvaquone resistance [14]. The current understanding of *T. evansi* genetic diversity at the P2 adenosine transporter-1 (*AT-1*) locus is limited, and potential links between mutations in P2 adenosine transporter-1 (*AT-1*) and diminazene resistance have yet to be demonstrated.

Low-throughput methods, including various types of PCRs, accompanied by Sanger sequencing, have been used for assessing genetic diversity [15]. However, these methods are more error-prone and expensive for large numbers of samples versus the high-throughput methods, such as deep amplicon sequencing (ADS) [16]. To this end, performing ADS by leveraging the Illumina Mi-Seq platform is a more reliable approach to identifying parasite amplicon sequence variants. This technique can generate sequence depth reads of up to 600 bp in length. Furthermore, primers that are barcoded can be pooled and can sequence up to 384 samples in a single run. This method has been used to study genetic diversity in the *Plasmodium* [17] and *Theileria* [18] species. Thus, the present study aimed to develop a high-throughput approach to exploiting ADS to improve the understanding of the *T. evansi* P2 adenosine transporter-1 (*AT-1*) locus.

## 2. Materials and Methods

### 2.1. Study Design, Field Sample, and Genomic DNA Extraction

A cross-sectional survey of *Trypanosoma* endemic regions in livestock was conducted in 7 areas, including Multan, Layyah, Rahim Yar Khan, Bahawalpur, Muzafargar, Lodhraan, and Dera Ghazi Khan of the Punjab province of Pakistan. The samples were collected between 2021 and 2023 from May to August during the peak transmission seasons of trypanosomiasis.

Twenty-six blood samples were collected from buffalo (n = 2), cattle (n = 14), camels (n = 7), goats (n = 1), and sheep (n = 2) presenting with clinical signs at veterinary hospitals. Once consent was obtained from the animal owners, the blood samples were collected by trained veterinarians. Briefly, the samples were collected by jugular venipuncture, and 5 mL of blood was withdrawn into EDTA tubes, which were further stored at −20 °C. Dr Morrison, Roslin Institute, University of Edinburgh, kindly provided *Trypanosoma*-negative control cattle blood samples. A total of 50 μL of blood was used for genomic DNA from each field sample following the protocols described in the TIANamp Blood DNA kit (Beijing Biopeony Co., Ltd., Beijing, China) [14]. Following this, a ’haemoprotobiome’ high-throughput sequencing methodology described in our previous study [19] was leveraged to confirm the *T. evansi* species from the collected samples (Appendix A).

### 2.2. Adapter and Barcoded PCR Amplification

A 555 bp fragment encompassing codons 178, 181, 239, and 286 of the *T. evansi* P2 adenosine transporter-1 (AT-1) locus was amplified with a de novo primer set (Appendix A). The overall scheme of the adapter and barcoded PCR approach is summarized in Figure 1 The de novo primer set was modified by adding adapters to each primer, allowing annealing. N indicated the number of random nucleotides between the primers and adapter sequence (Appendix A). Four forward (TeAT_For, TeAT_For-1N, TeAT_For-2N, and TeAT_For-3N) and four reverse primers (TeAT_Rev, TeAT_Rev-1N, TeAT_Rev-2N, and TeAT_Rev-3N) were mixed in equal proportions. The first-round PCR master mix included 1X KAPA HiFi Hot START Fidelity buffer, 10 mM dNTPs, 10 µM forward and reverse adapter primers, 0.5 U KAPA HiFi Hot START Fidelity Polymerase (KAPA Biosystems, Wilmington, Massachusetts, USA), 14.25 μL of ddH_2_O, and 5 μL of gDNA. The thermocycling conditions of the adapter PCR were: 95 °C for 2 min, followed by 35 cycles of 98 °C for 20 s, 62 °C for 15 s, 72 °C for 15 s, and a final extension of 72 °C for 5 min. The PCR products were then purified with AMPure XP Magnetic Beads (1X) (Beckman Coulter, Inc. California, United States) using a unique magnetic stand and plate (described by Yasein et al. (2022)) [19].

Finally, the barcoded PCR was performed using 8 forward and 12 reverse primer sets [14]. The barcoded PCR conditions were: 10 mM dNTPs, 1X KAPA HiFi Hot START Fidelity buffer, 14.25 μL of ddH_2_O, 0.5 U KAPA HiFi Hot START Fidelity Polymerase, and 2 μL of adapter PCR product as the DNA template. The barcoded forward (N501–N508) and reverse (N701-N712) primers (10 μM each) were obtained using Illumina Mi-Seq methods. The PCR thermocycling conditions were 98 °C for 45 s, followed by 7 cycles of 98 °C for 20 s, 63 °C for 20 s, and 72 °C for 2 min.

### 2.3. Amplicon Deep Sequencing

The schematic of the ADS approach using the Illumina Mi-Seq platform is shown in Figure 1 and described below. An amount of 10 μL of individually barcoded PCR products were combined to a pooled library and visualized/separated by agarose gel electrophoresis. The PCR products were then excised from the agarose gel using a commercial kit (QIAquick Gel Extraction Kit, Qiagen, Hilden, Germany), and 20 μL of the eluted DNA was further purified using AMPure XP magnetic beads (1X) to build a single purified DNA pooled library. The library was then measured by the KAPA qPCR library quantification kit (KAPA Biosystems, Wilmington, MA, USA). The library was then run on an Illumina Mi-Seq sequencer using a 600-cycle pair-end reagent kit (Mi-Seq Reagent Kits v2, MS-103-2003). The final library concentration was 15 nM with the addition of 15% Phix Control v3 (Illumina, FC-11-2003) [14,20].

### 2.4. Bioinformatic Analyses

For the post-run processing of Mi-Seq data, the platform used the barcoded indices to split all sequences by sample type and generate FASTQ files. FASTQ files were then analyzed by the Mothur v1.39.5 software [21,22], with slight modifications to the standard operating procedure (SOP) (Illumina Mi-Seq, available through the Mendeley database DOI: 10.17632/4cvwmvdgnj.1 Command Prompt pipeline). The overall bioinformatic analyses were described in our previous studies [14,18] and are shown in Figure 1. Briefly, the raw paired-end reads were run in the ‘make.contigs’ command for combining the two sets of reads (for each sample). By doing so, the sequences were extracted, and the quality score was assessed from the FASTQ files. This complemented the reverse and forward reads and then joined them into contigs. After this, the ambiguous sequence reads were removed, and the remaining data were aligned with the *T. evansi* P2 adenosine transporter-1 (*AT-1*) reference sequence library (through the Mendeley database DOI: 10.17632/4cvwmvdgnj.1), deploying the ‘align. seqs’ command. Only the samples yielding > 500 reads were included in the downstream analyses. The sequences that were not matched to the *T. evansi* P2 adenosine transporter-1 (*AT-1*) reference library were removed from the ‘summary. seqs’ command. Then, the *T. evansi* P2 adenosine transporter-1 (*AT-1*) sequence reads were further run on the ‘screen. seqs’ command to generate the FASTQ file. Finally, once the sequence reads were labeled as P2 adenosine transporter-1 (*AT-1*), a count list of the amplicon sequence variant of each isolate was created using the ‘unique. seqs’ command. The count list was then further used to create FASTQ files of the amplicon sequence variant of each isolate using the ‘split. groups’ command (Mendeley database DOI: 10.17632/4cvwmvdgnj.1).

### 2.5. Statistical Analyses of the Amplicon Variance

The amplicon sequence variants for the P2 adenosine transporter-1 (*AT-1*) locus were aligned using the MUSCLE alignment tool in the Geneious v9.1 software (Biomatters Ltd., Auckland Central, New Zealand) for the analyses of mutations associated with diminazene resistance at codons 178, 181, 239, and 286. The relative allele frequencies of *T. evansi* P2 adenosine transporter-1 (*AT-1*) resistance-associated mutations identified in the field isolates were estimated by dividing each isolate’s sequence reads by the total number of reads (R Core Team, 2013; package ggplot2). The genetic diversities of the P2 adenosine transporter-1 (*AT-1*) amplicon sequence variants were calculated among the isolates by using the DnaSP 5.10 software package [23]. The following values were obtained from the analyses: the number of segregating sites (S), heterozygosity (He), nucleotide diversity (π), and the mean number of pairwise differences (k).

## 3. Results

### 3.1. Genetic Diversity in the T. evansi P2 Adenosine Transporter-1 (AT-1) Locus

In total, 26 positive field samples [cattle (n = 14), buffalo (n = 2), camels (n = 7), sheep (n = 2), goats (n = 1), and negative control (n = 5)] were used to investigate genetic diversity in the *T. evansi* P2 adenosine transporter-1 (*AT-1*) locus. High levels of gene diversity were seen in the P2 adenosine transporter-1 (*AT-1*) locus of seven *T. evansi* isolates (Table 1) with potential resistance-type mutations (Table 2). The mean nucleotide diversity (π) of the seven isolates ranged from 0.013–0.041; segregating sites (S) ranged from 53–111; and pairwise differences (k) ranged from 7.36–9.90. Genetic diversity was comparatively low in seven *T. evansi* isolates (Table 1) with potential susceptible-type mutations (Table 2). The mean nucleotide diversities (π) of the seven isolates ranged from 0.003–0.009; segregating sites (S) ranged from 5–42; and pairwise differences (k) ranged from 3.24–8.53. Genetic diversity could not be calculated in twelve *T. evansi* isolates (Table 1) due to the presence of identical alleles with potential susceptible-type mutations (Table 2).

### 3.2. Potential Diminazene Resistance-Type Mutations

The presence of four potential diminazene resistance-type mutations at codons 178, 181, 239, and 286 of the P2 adenosine transporter-1 (AT-1) locus was determined by deep amplicon sequencing (Table 2). Diminazene resistance-type mutation GAA(178E) and TCA(178S) were detected in one isolate (Pop277). The GGT(239Y) resistance-type mutation was present in three isolates (Pop203, Pop278, and Pop287), GCT(239A) was present in one isolate (Pop266), and GCT(239A)/GAG(239E) was present in one isolate (Pop277) (Table 2). Diminazene resistance-type mutations AGC(286S), AGC(286H), ATC(286I), AAC(286N), and GAC(286D) were present in three isolates (Pop203, Pop207, and Pop279). The ATC(286I), GAC(286D), and ACC(286T) resistance-type mutations were present in three isolates (Pop266, Pop279, and Pop287) (Table 2). The GCA(A178), GGA(G181), GAT(D239), and AAC(N286) susceptible-type mutations were present in nineteen isolates (Table 2).

### 3.3. Allele Frequencies of Diminazene Resistance-Type Mutations

The frequencies of four diminazene resistance-type mutations at codons 178, 181, 239, and 286 of the P2 adenosine transporter-1 (AT-1) locus were determined in 26 *T. evansi* field isolates. The GAA(178E) and TCA(178S) resistance-type mutations were present at frequencies of 6.17% and 4.80% in one isolate (Pop277). The GAT(239Y), GCT(239A), and GTT(239E) resistance-type mutations were present at frequencies ranging between 5.59% and 14.26% in five isolates (Pop203, Pop266, Pop277, Pop278, and Pop287) (Table 3, Appendix A). The AGC(286S), AGC(286H), ATC(286I), GAC(286D), and ACC(286T) resistance-type mutations were present at frequencies ranging between 6.08% and 14.55% in six isolates (Pop203, Pop207, Pop266, Pop278, Pop279, and Pop 287) (Table 3, Appendix A). The GGA(G181), GCA(A178), GAT(D239), and AAC(N286) susceptible-type mutations were present at a frequency of 100% in nineteen isolates (Table 3, Appendix A).

## 4. Discussion

The current study used ADS for the first time to explore the genetic diversity of the *T. evansi* P2 adenosine transporter-1 (*AT-1*) locus and its potential link to diminazene resistance in endemic regions of Pakistan. These regions were chosen because animals are treated sporadically, often with generic diminazene drugs of unknown quality. Drug brands containing diminazene have been widely used throughout the globe to treat trypanosomiasis because they are relatively inexpensive and safe. Therefore, studying the genetic diversity in *T. evansi* species is essential to designing future strategies for the functional linkage of the P2 adenosine transporter-1 (*AT-1*) locus to diminazene resistance in this organism.

The genetic data demonstrate that the P2 adenosine transporter-1 (*AT-1*) locus of the *T. evansi* isolates is diverse. The high level of gene diversity combined with the high biotic potential of the parasite will inevitably confer genetic adaptability, eventually enabling the development of drug-resistance mutations. Livestock worldwide, including in Pakistan, are frequently treated with diminazene, and anecdotal evidence suggests that resistance is emerging (as per comm by Prof Kamran Ashraf). This worrisome situation implies that there is likely to be intense selection pressure for the development of drug resistance in the case of diaminazine. Furthermore, the underlying study has significant implications, considering that diminazene is among the few drugs available to treat trypanosomiasis in Pakistani livestock.

In the current study, we further reported the resistance-type mutations at codons 178E/178S, 239Y/239A/239E, and 286S/286H/286I/286D/286T of the P2 adenosine transporter-1 (*AT-1*) locus in *T. evansi* field isolates across all the seven cities of Pakistan. Our results demonstrated that 7 out of 26 *T. evansi* isolates had potential diminazene resistance-type mutations, with the allele frequencies ranging from 4.80–14.26%. In addition, contrary to the mutations found in this study, previous studies have documented diminazene resistance-associated P2 adenosine transporter-1 (*AT-1*) mutations at codons 178T, 239G, and 286S of *T. brucei gambiense* [8,9,10,11]. To this end, varying drug doses may explain the differences in these mutations in *T. evansi* and *T. brucei* and their observed frequency. Another highly speculative explanation may be that the mutations confer a fitness cost, depending on the parasites’ genetic background. A third explanation may be due to the genetic drift of susceptible- or resistant-type alleles with the movement of animals to new places. As there is a large amount of animal movement in the study region, gene flow may also play an essential role in spreading the infection and its associated mutation resistance. Finally, a fourth explanation is that the differences may be due to bottlenecking effects, which result in the loss of less common mutations [14].

## 5. Conclusions

To conclude, the current study is timely, as it reports a high level of genetic diversity among seven *T. evansi* field isolates that had mutations at codons 178E/S, 239Y/A/E, and 286S/H/I/D/T, which may play a significant role in developing a diminazene-resistant phenotype. The reported findings warrant further investigations to establish a mechanistic relationship between diminazene resistance and the mutations in P2 adenosine transporter-1 (*AT-1*).

## Figures and Tables

**Figure 1 animals-15-00756-f001:**
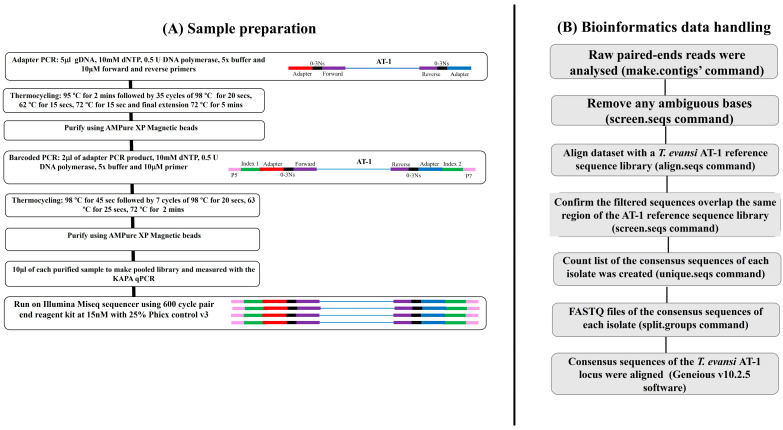
A flow diagram of sample preparation and bioinformatics data handling of the metabarcoded sequencing library.

**Table 1 animals-15-00756-t001:** Summary of genetic diversity indices for the P2 adenosine transporter-1 (AT-1) locus in 26 *T. evansi* isolates.

Field Isolates	Total No of Illumina Mi-Seq Reads	Susceptible-Type Reads	Resistance-Type Reads	Heterozygosity (He)	Nucleotide Diversity (π)	Segregating Sites (S)	Pairwise Differences (k)	Host	Endemic Region
Pop203	2675	1954	721	0.0011	0.01326	94	7.362	Camel	Rahim Yar Khan
Pop207	2261	1676	585	0.0014	0.01415	106	7.853	Camel	Rahim Yar Khan
Pop266	2716	1851	865	0.0031	0.01784	97	9.902	Camel	Multan
Pop277	2560	1771	508	0.0021	0.01587	111	8.806	Buffalo	Layyah
Pop278	1426	1103	323	0.0014	0.01408	101	7.812	Buffalo	Layyah
Pop279	1409	1230	179	0.0120	0.01464	53	8.123	Cattle	Layyah
Pop287	1954	1440	514	0.0150	0.04122	78	8.156	Goat	Rahim Yar Khan
Pop111	1262	1262		0.2720	0.00601	5	3.333	Cattle	Bahawalpur
Pop202	1397	1397		0.1770	0.00885	9	4.500	Cattle	Bahawalpur
Pop208	1346	1346		0.3140	0.00601	5	3.242	Cattle	Lodhraan
Pop210	2361	2361		0.0770	0.00955	15	3.000	Camel	Rahim Yar Khan
Pop268	2733	2733		0.0450	0.00529	21	5.600	Camel	Rahim Yar Khan
Pop283	1960	1960		0.0270	0.00367	42	8.538	Camel	Rahim Yar Khan
Pop284	1509	1509		0.0390	0.00461	37	8.109	Cattle	Layyah
Pop253	1708	1708		N/A				Sheep	Rahim Yar Khan
Pop199	1444	1444		N/A				Sheep	Rahim Yar Khan
Pop234	1614	1614		N/A				Cattle	Muzafargar
Pop217	2261	2261		N/A				Camel	Lodhraan
Pop220	2221	2221		N/A				Cattle	Bahawalpur
Pop237	1289	1289		N/A				Cattle	Dera Ghazi Khan
Pop230	2282	2282		N/A				Cattle	Dera Ghazi Khan
Pop238	1247	1247		N/A				Cattle	Bahawalpur
Pop25	2267	2267		N/A				Cattle	Dera Ghazi Khan
Pop114	2843	2843		N/A				Cattle	Dera Ghazi Khan
Pop100	3087	3087		N/A				Cattle	Muzafargar
Pop112	3276	3276		N/A				Cattle	Rahim Yar Khan

**Table 2 animals-15-00756-t002:** P2 adenosine transporter-1 (AT-1)-type mutations were detected in 26 *T. evansi* field isolates. Nonsynonymous mutations associated with resistance in *T. brucei* were identified at positions GAA(178E), TCA(178S), GGT(239Y), GCT(239A), GAG(239E), AGC(286S), CAC(286H), ATC(286I), GAC(286D), and ACC(286T). Other nonsynonymous mutations were identified at positions GCA(A178), GGA(G181), GAT(D239), and AAC(N286).

	Nucleotide	532–534 (GCA/GAA/TCA)	541–543 (GGA/GAA)	715–717 (GAT/GGT/GCT/GAG)	856–858 (AAC/AGC/CAC/ATC/GAC/ACC)	Host	Endemic Region
	Codon	178 A/E/S	181 G/E	239 D/Y/A/E	286 N/S/H/I/D/T		
Pop203		A	G	D/Y	N/S/H	Camel	Rahim Yar Khan
Pop207		A	G	D	N/H/I	Camel	Rahim Yar Khan
Pop266		A	G	D/A	N/I/D/T	Camel	Multan
Pop277		A/E/S	G	D/E/A	N	Buffalo	Layyah
Pop278		A	G	D/Y	N/S	Buffalo	Layyah
Pop279		A	G	D	N/D	Cattle	Layyah
Pop287		A	G	D/Y	N/T/D	Goat	Rahim Yar Khan
Pop100		A	G	D	N	Cattle	Bahawalpur
Pop111		A	G	D	N	Cattle	Bahawalpur
Pop112		A	G	D	N	Cattle	Lodhraan
Pop202		A	G	D	N	Camel	Rahim Yar Khan
Pop208		A	G	D	N	Camel	Rahim Yar Khan
Pop210		A	G	D	N	Camel	Rahim Yar Khan
Pop268		A	G	D	N	Cattle	Layyah
Pop283		A	G	D	N	Sheep	Rahim Yar Khan
Pop284		A	G	D	N	Sheep	Rahim Yar Khan
Pop253		A	G	D	N	Cattle	Muzafargar
Pop199		A	G	D	N	Camel	Lodhraan
Pop234		A	G	D	N	Cattle	Bahawalpur
Pop217		A	G	D	N	Cattle	Dera Ghazi Khan
Pop220		A	G	D	N	Cattle	Dera Ghazi Khan
Pop237		A	G	D	N	Cattle	Bahawalpur
Pop230		A	G	D	N	Cattle	Dera Ghazi Khan
Pop238		A	G	D	N	Cattle	Dera Ghazi Khan
Pop25		A	G	D	N	Cattle	Muzafargar
Pop114		A	G	D	N	Cattle	Rahim Yar Khan

**Table 3 animals-15-00756-t003:** Relative allele frequencies of the P2 adenosine transporter-1 (*AT-1*) resistance-type mutations in 7 *T. evansi* isolates. The relative allele frequency was based on the SNPs identified using deep amplicon sequencing technology.

Field Isolates	Susceptible Type Mutations %	Resistant Type Mutations %									
	GCA(A178) GGA(G181) GAT(D239) AAC(N286)	GAA(178E)	TCA(178S)	GGT(239Y)	GCT(239A)	GAG(239E)	AGC(286S)	CAC(286H)	ATC(286I)	GAC(286D)	ACC(286T)
Pop203	73.05			7.07			9.16	10.73			
Pop207	74.13							14.55	11.32		
Pop266	68.15				9.02				10.16	6.08	6.59
Pop277	69.18	6.17	4.80		5.59	14.26					
Pop278	77.35			10.17			12.48				
Pop279	87.30									12.70	
Pop287	73.69			10.18						8.50	7.63

## Data Availability

The following supporting information can be downloaded at: DOI: 10.17632/4cvwmvdgnj.1.

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
