# Peer review of "Genetic Diversity in the Diminazene Resistance-Associated P2 Adenosine Transporter-1 (AT-1) Gene of Trypanosoma evansi"

_animals, 2025, doi:10.3390/ani15050756_

Round 1
Reviewer 1 Report
Comments and Suggestions for Authors
Report on “Genetic diversity in diminazene resistance-associated P2 adenosine transporter-1 (AT-1) gene of Trypanosoma evansi” by Ali et al for Animals.
This is an interesting report on the genetic diversity in the T. evansi Adenosine Transporter 1 (TevAT1) gene in a set of isolates from domestic animals in Pakistan. The work is important for the understanding of the prevalence of drug resistance markers in this pathogen population and has been competently executed. I have no remarks on the technical side of things. To the extent I am competent in these techniques, the approach seems reasonable and the results reliable. However, I do have some suggestions about the contextualisation and implications, and would like to suggest some additional background references.
Specific points:
- I acknowledge that diminazene is the most common treatment for animal trypanosomiasis used in Pakistan but the authors correctly state that it is not the only one in clinical use. I would like a bit more information and context on the alternatives. Camels with surra are more often treated with suramin than with diminazene, in most countries, and horses are sometimes treated with cymelarsan or quinapyramine, for instance. This is important to mention because mutations in AT1 in T. evansi and T. brucei can also give (partial) resistance to the arsenical drugs including cymelarsan (e.g. Suswam et al 2001, doi: 1016/s0304-4017(01)00533-7). Even if that drug is rarely used locally in this area of Pakistan, it would be important to mention it. There is no known association between the AT1/P2 transporter and resistance to suramin or quinapyramine.
- The classification into ‘resistance-type mutations’ and ‘susceptibility-type mutations’ is somewhat troublesome, and as it is, it will not be understood by most readers, I believe. First, the term ‘susceptibility mutations’ is confusing. In my terminology these are the ‘wild-type’ or ‘consensus’ sequences, not mutations, which literally means a change from something, but these are the original codons, so I suggest that the authors clarify that and ideally change the designation as susceptibility mutations. Secondly, the term ‘resistance mutation’ or ‘resistance-type mutation’ implies strongly that the presence of this one SNP will result in resistance to diminazene but this is not being tested. In fact, I would think that it is unlikely that any and all amino acid changes at codons 178, 181, 239 and 286 confer resistance. The authors do not quote or discuss the one pertinent paper in this matter, Munday et al, Molecular Microbiology 2015 (doi: 1111/mmi.12979) where all the known resistance mutations were individually introduced and expressed in vitro, followed by characterisation; several combinations of SNPs were also tested in the same way. They found that none of the SNPs in the reported ‘resistance allele’, which contained multiple mutations form the WT, conferred a significant level of diminazene resistance, but that combinations of multiple of the SNPs did. This information should be mentioned and discussed appropriately. They should also mention that it is very unlikely that a heterozygote with one mutant (resistance) allele and one WT (susceptibility) allele will be diminazene susceptible. To the best of my knowledge, the reports of resistance alleles, e.g. from the groups of pascal Maser or Enock Matovu, were always homozygous, and theoretically the presence of fully functional P2 transporters would of course allow the entry of diminazene regardless of the presence of some with disabling SNPs.
- The authors should, in the Discussion, mention the possibility of a fitness cost. Resistance alleles in b. gambiense AT1 disappeared from a population in an area in Uganda where selective pressure was removed, but not from another area in Uganda where treatments with arsenicals (melarsoprol) continued (see Pyana Pati et al., PLOS Negl Trop Dis 2014; Kazibwe et al., PLOS NTD 2009, from the groups of Phillippe Buscher and Enock Matovu, respectively. This is one indication that there is cost in fitness or transmission to the resistance alleles if, possibly, not to (all of) the individual SNPs.
- Minor issue. 3rd line of Introduction: b. brucei is not the main infectious agent causing nagana in sub-Saharan Africa, as T. vivax and T. congolense have a higher prevalence (Okello et al. J Med Entomol 2022 (DOI: 10.1093/jme/tjac018).
- Introduction, 2nd paragraph: “The P2 adenosine transporter-1 (AT-1) gene is needed to uptake diminazene in Trypanosoma ” reference 5 (Barrett et al., 1995) is certainly appropriate for T. equiperdum but Witola et al Exp Parasitol 2004 could be added for T. evansi. The full pharmacology of diminazene against various parasite species including T. brucei and T. congolense is described by Andrew S. Peregrine and Mohammed Mamman (Acta Tropica 54, 185-203 (1995), DOI: 10.1016/0001-706x(93)90092-p).
- Introduction, 3rd paragraph: reference 7 from Delespaux is based on a discredited assumption, assuming the existence of an AT1/P2 gene in T. congolense. Munday et al 2013 have shown that is not the case [10.1016/j.ijpddr.2013.01.004]. Although I do not require or expect the authors to address this issue, I do suggest that instead of reference 7, a more appropriate reference is from the Maser group: Graf et al., PLOS Negl Trop Dis 2013 (doi: 10.1371/journal.pntd.0002475).
- Results, section 3.2: “resistance-type mutations 209 AGC(286S)/AGC(286H)” typo in the codon for histidine, as AGC is coding for serine. The authors may want to check for similar typos.
Author Response
Please see the attached file for reviewer comments

Reviewer 2 Report
Comments and Suggestions for Authors
The emergence of drug resistance is a global problem for all parasites and microbes. So, the current study was designed to confirm the diminazine resistance of T. evansi by application of deep amplicon. The main shortage of this paper is the samples, and it was not clear if the authors examined a large number of animals or how they obtained the samples of cattle, camels, buffaloes, sheep, and goats. Also, the authors used a control negative or susceptible isolate of T. evansi, but the results didn't include it. Why didn't the authors use control positive of T. bruceii? Another issue is that genetic diversity is common in T. evansi so finding mutations is not a surprise; the authors should discuss this point. Where is the conclusion? lastly, the semilarity index is very high (44%).

Author Response
No comments of Reviewer 2
Round 2
Reviewer 2 Report
Comments and Suggestions for Authors
The authors commented sufficiently to all concerns
Comments on the Quality of English LanguageThe English language in the manuscript is fine from my view